# Management Options for Fetal Bronchopulmonary Sequestration

**DOI:** 10.3390/jcm11061724

**Published:** 2022-03-20

**Authors:** Magdalena Litwinska, Ewelina Litwinska, Krzysztof Szaflik, Marzena Debska, Tomasz Szajner, Katarzyna Janiak, Piotr Kaczmarek, Miroslaw Wielgos

**Affiliations:** 1Department of Obstetrics and Gynecology, Medical University of Warsaw, 02-091 Warszawa, Poland; ewelina.litwinska@gmail.com (E.L.); marzena@debska.me (M.D.); miroslaw.wielgos@gmail.com (M.W.); 2Department of Gynecology, Fertility and Fetal Therapy, Polish Mother’s Memorial Hospital—Research Institute, 93-338 Lodz, Poland; krzysztofszaflik@wp.pl (K.S.); kasiajaniak@me.com (K.J.); kaczmarekpiotr1@gmail.com (P.K.); 3Department of Obstetrics and Gynecology, Pro-Familia Hospital, 35-001 Rzeszów, Poland; tomissek@gmail.com

**Keywords:** bronchopulmonary sequestration, fetal therapy, thoraco-amniotic shunt, laser coagulation of the feeding vessel

## Abstract

To evaluate the prenatal course and perinatal outcome of fetuses with bronchopulmonary sequestration (BPS) managed expectantly or using minimally invasive methods. This was a retrospective study of 29 fetuses with suspected BPS managed between 2010 and 2021 in three fetal medicine centers in Poland. Medline was searched to identify cases of BPS managed expectantly or through minimally-invasive methods. In 16 fetuses with BPS, there was no evidence of cardiac compromise. These fetuses were managed expectantly. Thirteen hydropic fetuses with BPS qualified for intrauterine intervention: a thoraco-amniotic shunt (TAS) was inserted in five fetuses, laser coagulation of the feeding vessel was performed in seven cases, and one fetus had combined treatment. In the combined data from the previous and the current study of various percutaneous interventions for BPS associated with hydrops, the survival rate was 91.2% (31/34) for TAS, 98.1% (53/54) for laser coagulation, and 75% (3/4) for intratumor injection of sclerosant. After taking into account cases with available data, the rate of preterm birth before 37 weeks in the group treated with laser coagulation was 14.3% (7/49) compared to 84.6% (22/26) in the group treated with TAS. The need for postnatal sequestrectomy was lower in the group of fetuses treated with laser coagulation 23.5% (12/51) in comparison to fetuses treated with TAS 84% (21/26). In fetuses with BPS without hydrops, progression of the lesion’s volume, leading to cardiac compromise, is unlikely. In hydropic fetuses with BPS, intrauterine therapy using minimally invasive methods prevents fetal demise. Both, the rate of preterm birth and the need for postnatal surgery is significantly lower in the group treated with laser coagulation compared to the group treated with TAS.

## 1. Introduction

Bronchopulmonary sequestration (BPS) is a rare disorder of the lower respiratory tract, presenting as a solid lesion receiving blood supply from the systemic artery. The prognosis of this condition is generally favorable unless there is associated hydrothorax or hydrops which are thought to be the consequences of impaired cardiac function due to mediastinal shift and compression of systemic veins [1,2,3,4,5,6].

In the case of large lesions with hydrops, several attempts at percutaneous fetal intervention have been described, with the aim of improving perinatal outcomes. These include: placement of thoraco-amniotic shunt [1,2,7,8,9,10,11,12,13,14,15,16,17,18,19], thoracocentesis [7,8,14,16,18], laser coagulation of the feeding vessel [17,20,21,22,23,24,25,26,27,28], intratumor injection of sclerosant [29,30] and combined treatment [28,31]. Although the condition is rare and therefore the reported data are limited, it is evident that invasive management in hydropic fetuses is beneficial.

The objective of this study is firstly, to report our experience with the management of 29 fetuses diagnosed with BPS with and without associated hydrothorax and/or hydrops and secondly, to review the literature on this kind of pathology.

## 2. Materials and Methods

This was a retrospective study of 29 fetuses diagnosed with bronchopulmonary sequestration in three fetal medicine centers in Poland in the years 2006–2021. Within this period, 289 fetuses with various echogenic lung lesions were referred for further diagnosis and management. In this group, 34 fetuses were suspected of BPS, but five fetuses were excluded from the study because of additional abnormalities. Out of the 29 fetuses with BPS, thirteen met the qualification criteria for fetal therapy, firstly, because of the presence of a large solid lesion receiving blood supply from a clearly identifiable vessel originating from the aorta, secondly, due to the presence of hydrothorax and/or hydrops, and thirdly, because of the absence of other major defects.

Preoperatively, a detailed ultrasound examination was carried out to exclude any other major defects and to determine the presence and severity of pleural effusion and hydrops, as well as to measure the CCAM Volume Ratio (CVR). Essentially, the volume of the lesion was calculated for the maximum transverse, anteroposterior, and longitudinal diameter and then divided by the head circumference [CVR = (length × width × height of the lesion × 0.52)/head circumference]. Specialist fetal echocardiography was also carried out to exclude cardiac defects, assess the size of the heart and muscle contractility, diagnose possible atrioventricular regurgitation, and evaluate the blood flow using Doppler techniques.

All patients received detailed counseling by a fetal medicine specialist and a pediatric surgeon concerning the nature of the lesion and likely prognosis. In the case of fetuses that met the qualification criteria for intrauterine therapy, a written consent form for thoraco-amniotic shunting or laser coagulation of the feeding vessel was taken, after counseling about possible benefits, risks, and complications from this kind of treatment.

Up to January 2013, cases of BPS with hydrops/hydrothorax were treated with thoraco-amniotic shunts. Thereafter, they were treated with laser coagulation of the feeding vessel. Thoraco-amniotic shunt insertion was conducted using the technique first described by Rodeck et al. [2]. Ultrasound scanning was used to obtain a transverse section of the fetal thorax and to define the appropriate site of entry on the maternal abdomen which was infiltrated with local anesthetic (10 mL of 10% lignocaine) down to the myometrium. Under continuous ultrasound guidance, a metal cannula with a trochar (external diameter 3 mm, length 15 cm; Rocket KCH Reusable Introducer Set Washington, United Kingdom) was introduced transabdominally into the amniotic cavity and inserted through the fetal chest wall into the pleural cavity. The trochar was then removed and the shunt (diameter 2 mm, length 12 cm; Rocket KCH Fetal Bladder Catheter, Washington, UK) was inserted into the cannula. A short introducer rod was then used to insert half of the catheter into the pleural cavity. Subsequently, the cannula was gradually removed into the amniotic cavity where the other half of the catheter was pushed by a longer introducer. Laser coagulation of the feeding vessel was conducted under ultrasound guidance. The cross-section of the fetal thorax was visualized using ultrasound scanning and an 18-G needle was introduced through the fetal thorax. A 0.7 mm laser fiber was then inserted through the needle with its tip pointing directly at the feeding artery. The vessel was coagulated for 6–12 s using the output of 40–50 W. The procedure was repeated three times until the absence of blood flow using color Doppler was confirmed. In cases of polyhydramnios (amniotic fluid index > 25 cm), amniodrainage was carried out through the cannula/needle. Perioperative tocolysis was provided by betamimetic if the gestation was ≥24 weeks. All patients received antibiotic prophylaxis (Penicillin 1.2 g IV).

In fetuses that qualified for expectant management, serial ultrasound scans every two weeks were offered. Fetuses that were managed with minimally-invasive methods were followed every day for the first week and every 1–2 weeks thereafter until delivery, to confirm the resolution of hydrops and the lack of its recurrence. After delivery, the chest drains were immediately clamped or removed to avoid the development of pneumothorax.

The qualification criteria for postnatal surgery included superadded infection presenting as a lung abscess, recurrent pneumonia and empyema, increasing lesion volume-limiting respiratory system, and lesions suspected of malignancy. In participating centers, the preferred method of treatment is using minimally invasive thoracoscopic excision which is preferably performed before one year to prevent the vast majority of complications.

The perinatal data of the fetuses that underwent the intrauterine procedures were obtained from the database of participating centers. The fetal and newborn characteristics included in the analysis were intrauterine death, gestational age, and postnatal follow-up including surgery.

### 2.1. Literature Search

Searches of Medline and Embase were performed to identify all studies in the English language that reported on the expectant and invasive management of fetuses diagnosed with BPS. In the case of fetuses managed expectantly, only reports of at least two fetuses were considered [1,10,17,18,19,22,32,33,34,35,36,37,38,39,40,41,42,43].

### 2.2. Statistical Analysis

The need for postnatal surgery and the rate of preterm birth before 37 weeks were compared between the groups treated with TAS and with laser coagulation of the feeding vessel, using the Chi-square test with Yates correction.

## 3. Results

The characteristics of the 29 fetuses with BPS included in our study are summarized in Table 1 (expectant management) and Table 2 (minimally-invasive treatment).

Fetuses managed expectantly

The median gestational age at the time of initial diagnosis was 22 (range 20–28) weeks. 

In six cases, thoracic masses were described as extralobar, receiving the blood supply from the thoracic descending aorta (TDA) and abdominal descending aorta (ADA) in four and two cases, respectively. In the remaining ten cases, thoracic masses were described as intralobar, receiving the blood supply from TDA. In five cases, BPS was associated with a major mediastinal shift. There were no cases of pleural effusion or hydrops in this group. In one case there was mild polyhydramnios. In the prenatal ultrasound examinations, no additional abnormalities were detected and in all cases the fetal karyotype was normal.

The partial regression, no change, and progression of the BPS volume in the subsequent ultrasound scans were found in ten, four, and two cases, respectively. The median gestational age at birth was 39 (37–40) weeks and in all cases, the 5-min Apgar score was >8. Postnatal excision of the lesion was carried out in four neonates. All 16 survivors had follow-up for at least 12 months after and they are healthy with no neurological disabilities.

Fetuses managed with intrauterine therapy

The median gestational age at the time of intervention was 23 (range 20–27) weeks. In all cases, the thoracic mass of extralobar type with blood supply from the thoracic descending aorta was associated with a major mediastinal shift. In all cases, there were associated hydrops and polyhydramnios. No additional abnormalities were detected, and in all cases, the fetal karyotype was normal.

In the group of fetuses treated with a thoraco-amniotic shunt, the procedure was conducted successfully in all cases and resulted in the resolution of hydrops within one week of shunt insertion. However, in one case the procedure was repeated because of the dislocation of the initial shunt into the amniotic cavity, and hydrops recurrence. No significant change in the volume of BPS was found in subsequent ultrasound scans. Four fetuses were delivered vaginally before 37 weeks due to the preterm rupture of membranes. One fetus was born via elective cesarean section at 38 weeks (breech presentation).

In the group of fetuses treated with laser coagulation of the feeding vessel, the procedure was uncomplicated in all cases, and in seven out of eight fetuses, resulted in the resolution of the hydrops within one week. Also, significant regression of the thoracic mass was found in the subsequent ultrasound scans. In one case, laser coagulation successfully ceased the blood supply, however, hydrops had not resolved within 10 days after the procedure and the patient was offered another intervention using a thoraco-amniotic shunt which was successfully inserted at 22 weeks gestation, leading to complete resolution of hydrops. Postnatal computed tomography showed two aberrant arteries supplying the lesion which were not identified prenatally. This unusual pattern of vascularization was presumably the reason why laser coagulation did not bring expected results at the first attempt. All fetuses were born at term; five did not require surgery and three had sequestrectomy due to an increased risk of pulmonary infections.

The literature search identified a total of 244 fetuses managed expectantly, and 96 fetuses managed with minimally-invasive methods (thoraco-amniotic shunt, laser coagulation of the feeding vessel, alcohol injection) between 1986 and 2021 (Appendix A). In the combined data from the previous and the current study, the survival rate was 98.3% in the expectant management group (93.9% non-hydropic fetuses) and 94.8% in the invasive management group (100% hydropic fetuses). The need for postnatal surgery in the expectant management group was 63.3%. In the invasive management group, the need for postnatal sequestrectomy was significantly lower in fetuses treated with laser coagulation of the feeding vessel 23.5% (12/51) in comparison to fetuses treated with thoraco-amniotic shunts 84% (21/26) [*p* < 0.0001]. In the group treated with laser coagulation, the rate of preterm birth (PTD) before 37 weeks was 16.7% and in the group treated with a thoraco-amniotic shunt, the rate of PTD was 63.6% [*p* < 0.0001] [Appendix A].

## 4. Discussion

Main findings of this study

The data from this study and previous reports demonstrate the favorable prognosis in cases of BPS without associated hydrothorax and/or hydrops. It also shows the efficiency of intrauterine therapy in the treatment of BPS with associated hydrops. Moreover, laser coagulation of the feeding vessel is likely to contribute to the reduction of the lesion’s volume and as a result, the need for postnatal surgery is diminished.

In the expectant management group, a lack of progression of the lesion’s volume was found in the vast majority of cases (87.5%). All infants were born in a good overall condition and did not require any treatment in the first month of life. In the participating centers the inclusion criteria for fetal intervention in the case of BPS are either hydrops or severe hydrothorax. Additional abnormalities were found in 14.7% of cases. These cases were excluded from further analysis.

In all fetuses with hydrops, the intrauterine management using the thoraco-amniotic shunt or laser coagulation of the feeding vessel, resulted in the resolution of hydrops. There were no cases of therapy complications. Additionally, in two fetuses treated with the thoraco-amniotic shunt, there was a need for reinsertion because the first one became dislodged. The intrauterine management using thoraco-amniotic shunt resulted in the resolution of fetal hydrops without an impact on the lesion’s size.

Limitations of the study

The major limitations of the study relate to its retrospective design, the small size of the group, and the lack of controls. However, these limitations result from the rarity of the condition.

Comparison of the findings with previous studies in the literature

Thoraco-amniotic shunt insertion in a hydropic fetus with bronchopulmonary sequestration was first described by Weiner et al. [7]. Treatment resulted in the resolution of hydrops, however, the fetus was delivered prematurely. Another approach to the treatment of BPS by interstitial laser was first reported by Oepkes et al. in a fetus at 23 weeks gestation presenting with a large solid mass and hydrops [20]. Cavoretto et al. were the first to publish a series of cases treated with laser ablation and proved the feasibility and safety of this technique [22]. Since then a number of studies have been published. In a large series by Mallman et al. [17], that compared thoraco-amniotic shunting and interstitial laser in BPS with pleural effusion, the survival rate was 85.7% (6/7) in the group treated with thoraco-amniotic shunting and 100% (5/5) in the group treated with laser coagulation. However, the authors found that laser ablation was associated with significantly better perinatal outcome compared to the group treated with thoraco-amniotic shunting: gestational age at delivery was higher (median age, 39.1 (range 38.0–40.0) vs. 37.2 (range 30.3–37.4)) and the need for postnatal sequestrectomy was lower (20% vs. 83.3%). In this extended summary, including our data, the survival rate in the invasive management group was 97.1%: 93.7% (30/32) in the group treated with thoraco-amniotic shunting, and 98% (50/51) in the group treated with laser coagulation. However, survival does not always mean success, defined as the term birth of a neurologically healthy child. The major risk factor of neurological impairment is still preterm delivery. 

In the group treated with laser coagulation, the rate of PTD before 37 weeks was 14.3% (7/49) and in the group treated with thoraco-amniotic shunting, the rate of PTD was 84.6% (22/26). Also, the need for postnatal sequestrectomy was lower in the group of fetuses treated with laser coagulation 12/51 (23.5%) in comparison to fetuses treated by thoraco-amniotic shunts 21/26 (84%).

Implications for clinical practice

The role of intrauterine therapy in cases of hydropic fetuses with bronchopulmonary sequestration is well accepted. Several case reports suggest that fetuses with BPS and hydrops managed expectantly, have a very poor prognosis due to pulmonary hypoplasia [34,35]. Successful treatment with the thoraco-amniotic shunt contributes to the improvement of fetal well-being and diminishes the likelihood of intrauterine demise as well as neonatal death due to pulmonary hypoplasia. However, this kind of treatment is strictly symptomatic. Laser coagulation of the feeding vessel is likely to not only diminish the signs of hydrops but also to be the definitive treatment by halting the blood supply to the pathological lung mass.

It should therefore be considered as the most appropriate method of treatment in hydropic fetuses with BPS.

## Figures and Tables

**Table 1 jcm-11-01724-t001:** Data of fetuses with BPS managed expectantly.

Case	Ultrasound Findings at the Time of Initial Diagnosis							Prenatal Course	Follow Up				
	GA	Side	Type	Vessel	CVR	AFI	Mediastinal Shift/Hydrops		Outcome	Birth	Apgar	Age at Surgery	Reason for Surgery
1	21	Left	Intralobar	TDA	1.22	18	+/−	Regression	LB	38	9	No intervention	
2	24	Left	Extralobar	TDA	0.43	12	−/−	Regression	LB	39	9	No intervention	
3	22	Right	Extralobar	TDA	1.02	11	−/−	Regression	LB	38	9	No intervention	
4	21	Left	Intralobar	TDA	0.67	13	−/−	Progression	LB	38	8	12	Risk of heart failure
5	23	Left	Extralobar	ADA	0.34	12	−/−	No change	LB	40	10	No intervention	
6	24	Left	Intralobar	TDA	0.39	14	−/−	Regression	LB	37	8/9	7	Risk of recurrent infections
7	23	Left	Intralobar	TDA	0.98	12	−/−	Regression	LB	40	9	No intervention	
8	21	Left	Extralobar	TDA	1.12	12	−/−	No change	LB	39	9	12	Parental anxiety
9	20	Right	Extralobar	ADA	1.13	25	+/−	Regression	LB	37	8	No intervention	
10	21	Right	Intralobar	TDA	1.25	18	+/−	No change	LB	39	9	7	Risk of recurrent infections
11	28	Left	Intralobar	TDA	1.34	19	+/−	Regression	LB	40	10	No intervention	
12	22	Left	Intralobar	TDA	1.12	14	−/−	Regression	LB	38	9	No intervention	
13	24	Right	Extralobar	TDA	0.98	18	−/−	Regression	LB	36	10	No intervention	
14	20	Left	Intralobar	TDA	0.77	21	−/−	No change	LB	39	10	No intervention	
15	26	Left	Intralobar	TDA	1.21	23	+/−	Regression	LB	40	9	No intervention	
16	22	Left	Intralobar	TDA	0.55	18	−/−	Progression	LB	36	10	No intervention	

GA = gestational age in weeks; Vessel = origin of the feeding vessel; TDA = Thoracic Descending Aorta; ADA = Abdominal Descending Aorta; AFI = Amniotic Fluid Index; LB = livebirth; Birth = gestational age at birth in weeks; Surgery = age at postnatal surgery in months.

**Table 2 jcm-11-01724-t002:** Data of fetuses with BPS managed with intrauterine intervention.

Case	Findings at the Time of Surgery							Type of Intervention		Follow Up				
	GA	Side	Type	Vessel	CVR	AFI	Mediastinal Shift/Hydrops	Shunt (no.)	Laser	Outcome	Birth	Apgar	Surgery	Reason for Surgery
1	21	Right	Extralobar	TDA	1.49	28	yes	Shunt × 2		LB	36	9	13	Risk of infections
2	23	Left	Extralobar	TDA	2.01	30	yes	Shunt × 1		LB	34	8	12	Risk of heart failure
3	20	Left	Extralobar	TDA	1.62	26	yes	Shunt × 2		LB	38	9	10	Risk of heart failure
4	24	Left	Extralobar	TDA	2.10	25	yes	Shunt × 1		LB	35	9	12	
5	25	Right	Extralobar	TDA	1.9	27	yes	Shunt × 1		LB	34	8	2	
6	20	Left	Extralobar	TDA	1.56	28	yes	Shunt × 1	Laser	LB	40	10	6	Risk of infections
7	23	Left	Extralobar	TDA	1.79	24	yes		Laser	LB	39	9	2	Risk of infections
8	21	Left	Extralobar	TDA	1.42	24	yes		Laser	LB	39	10	No intervention	
9	27	Right	Extralobar	TDA	2.88	28	yes		Laser	LB	41	10	No intervention	
10	25	Right	Extralobar	TDA	3.36	31	yes		Laser	LB	40	9	No intervention	
11	23	Left	Extralobar	TDA	2.45	27	yes		Laser	LB	39	9	No intervention	
12	22	Right	Extralobar	TDA	2.15	25	yes		Laser	LB	40	10	12	Parental anxiety
13	22	Left	Extralobar	TDA	1.95	19	yes		Laser	LB	39	10	No intervention	

GA = gestational age in weeks; TDA = Thoracic Descending Aorta; AFI = Amniotic Fluid Index; CVR = congenital pulmonary airway malformation volume ratio; LB = livebirth; Birth = gestational age at birth in weeks; Surgery = age at postnatal surgery in months.

## Data Availability

The datasets used and analyzed during the current study are available from the corresponding author on reasonable request.

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
