# Peer review of "Management Options for Fetal Bronchopulmonary Sequestration"

_jcm, 2022, doi:10.3390/jcm11061724_

Round 1
Reviewer 1 Report
A study with a retrospective cohort of cases of bronchopulmonary sequestration is presented.
It is one of the largest cohorts presented so far in the literature.
The study is well presented and the results are consistent with the objectives.
As only comments:
- Perhaps the table Tables 3 or 4, review of the literature, occupies many pages and perhaps I would pass it on as complementary material and describe the most outstanding in the text.
- In the conclusions it is proclaimed that the treatment of choice could be laser, but in this study no statistical calculation has been made to compare the two treatments. It would be nice to have the information of the statistical comparison.
Thanks
Author Response
Reviewer 1 response:
A study with a retrospective cohort of cases of bronchopulmonary sequestration is presented.
It is one of the largest cohorts presented so far in the literature.
The study is well presented and the results are consistent with the objectives.
Response: Thank you.
As only comments:
- Perhaps the table Tables 3 or 4, review of the literature, occupies many pages and perhaps I would pass it on as complementary material and describe the most outstanding in the text.
Response: Thank you. I agree that Tables 3 and 4 could be moved to complementary material. I have now added a comment on the literature search in the last paragraph of Results (page 6).
- In the conclusions it is proclaimed that the treatment of choice could be laser, but in this study no statistical calculation has been made to compare the two treatments. It would be nice to have the information of the statistical comparison.
Response: Thank you. Statistical calculation has now been added in the Methods (paragraph: statistical analysis) and in the Results (page 6).
Reviewer 2 Report
Reviewer Comments
This is a large multicenter study. The authors reported their own experience on management of cases with bronchopulmonary sequestration (BPS) and they combined the existing literature on the current management options of this rare condition. The study demonstrated that first, cases with BPS without hydrops has generally good prognosis with all infants being born alive in good condition with no significant increase in the lesion`s volume and second cases treated with minimally invasive methods are associated with major reduction of fetal demise. The study demonstrated that the use of laser coagulation of the feeding vessel versus thoraco-amniotic shunting in BPS is associated with lower rate of postnatal sequestrectomy 23.5% versus 84% and also reduction in the rate of premature delivery before 37 weeks of 16.7% versus 63.6%.
Introduction
The introduction is clear.
Patients and Methods
The cases included from the three centers in Poland have informed consents and the study is designed appropriately.
The authors should pay attention to line 115 in the manuscript to clarify the terminology, they previously used in the text “fetuses”, where here they use “foetuses”, please replace with the correct one.
Results
The information provided in the subsequent tables (1-4) is well described and extracted correctly from the current literature reporting on management of BPS.
The cases reported from their own series in terms of management options expectant and minimally invasive in cases with non-hydrops vs hydrops are compatible with the existing literature.
Discussion
The discussion is clear.
This is an excellent work. The manuscript is very well written and is easy to read, the methodology is well described, the results are clearly exposed, and the findings well discussed, including limitations of the study. I have no criticism to this manuscript.
Author Response
This is a large multicenter study. The authors reported their own experience on management of cases with bronchopulmonary sequestration (BPS) and they combined the existing literature on the current management options of this rare condition. The study demonstrated that first, cases with BPS without hydrops has generally good prognosis with all infants being born alive in good condition with no significant increase in the lesion`s volume and second cases treated with minimally invasive methods are associated with major reduction of fetal demise. The study demonstrated that the use of laser coagulation of the feeding vessel versus thoraco-amniotic shunting in BPS is associated with lower rate of postnatal sequestrectomy 23.5% versus 84% and also reduction in the rate of premature delivery before 37 weeks of 16.7% versus 63.6%.
Introduction
The introduction is clear.
Patients and Methods
The cases included from the three centers in Poland have informed consents and the study is designed appropriately.
The authors should pay attention to line 115 in the manuscript to clarify the terminology, they previously used in the text “fetuses”, where here they use “foetuses”, please replace with the correct one.
Results
The information provided in the subsequent tables (1-4) is well described and extracted correctly from the current literature reporting on management of BPS.
The cases reported from their own series in terms of management options expectant and minimally invasive in cases with non-hydrops vs hydrops are compatible with the existing literature.
Discussion
The discussion is clear.
This is an excellent work. The manuscript is very well written and is easy to read, the methodology is well described, the results are clearly exposed, and the findings well discussed, including limitations of the study. I have no criticism to this manuscript.
Response: Thank you. Spelling mistakes have now been corrected.